# Generating Datasets of 3D Garments with Sewing Patterns

**Maria Korosteleva**
Graduate School of Culture Technology
KAIST
mariako@kaist.ac.kr

**Sung-Hee Lee**
Graduate School of Culture Technology
KAIST
sunghee.lee@kaist.ac.kr

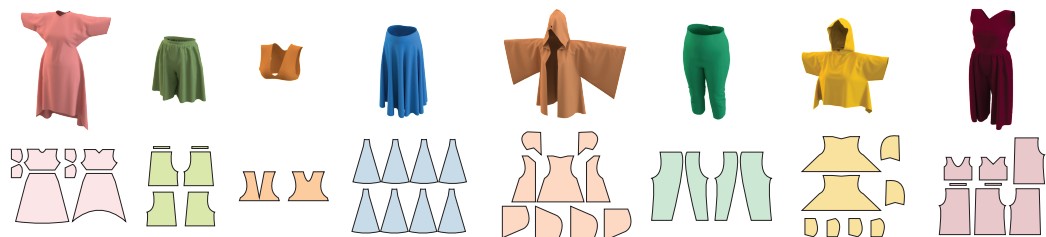

Figure 1: Example garments from our dataset of 3D Garments with Sewing patterns. The dataset is obtained with our original garment generation pipeline that samples novel designs in sewing patterns space.

## Abstract

Garments are ubiquitous in both real and many of the virtual worlds. They are highly deformable objects, exhibit an immense variety of designs and shapes, and yet, most garments are created from a set of regularly shaped flat pieces. Exploration of garment structure presents a peculiar case for an object structure estimation task and might prove useful for downstream tasks of neural 3D garment modeling and reconstruction by providing strong prior on garment shapes. To facilitate research in these directions, we propose a method for generating large synthetic datasets of 3D garment designs and their sewing patterns. Our method consists of a flexible description structure for specifying parametric sewing pattern templates and the automatic generation pipeline to produce garment 3D models with little-to-none manual intervention. To add realism, the pipeline additionally creates corrupted versions of the final meshes that imitate artifacts of 3D scanning.

With this pipeline, we created the first large-scale synthetic dataset of 3D garment models with their sewing patterns [1]. The dataset contains more than 20000 garment design variations produced from 19 different base types. Seven of these garment types are specifically designed to target evaluation of the generalization across garment sewing pattern topologies.

## 1 Introduction

The task of learning-based object structure estimation from various input modalities has seen significant development in recent years. Both reconstruction and generative models that rely on structure tend to produce more consistent and plausible results comparing to the models without explicit structure modeling, as demonstrated by the recent successes in the area [2, 3, 4, 5]. Yet, most of the works in the domain rely on the datasets of rigid objects, while datasets of structured deformable objects suitable for Deep Learning applications are practically not available.

35th Conference on Neural Information Processing Systems (NeurIPS 2021) Track on Datasets and Benchmarks.

On the other hand, the growing areas of 3D garments reconstruction and learning-based cloth simulation [6, 7, 8, 9, 10] do target deformable objects but rarely exploit the fact that most garments have a well-established base structure – a sewing pattern. The 3D garment data in these domains are scarce, and no known datasets of significant size provide the garment sewing patterns alongside the 3D garment models or renders. Another challenge is presented by the variety of designs the garments exhibit in the real world, and existing datasets only describe a fraction of the design space.

To address these issues, we propose a flexible generation pipeline that allows dense sampling from the design space of garment sewing patterns. We assume a *sewing pattern* to be a collection of the 2D pieces of fabric (panels) with a known placement of each panel around the human body and information of how the panels are stitched together to form the final garment. We model a *panel* to be a closed piece-wise curve with every piece (*edge*) being either a straight line or a Bezier spline. Such a sewing pattern is a close approximation of how most real-world garments are designed.

The first step of our pipeline is the definition of a parametric sewing pattern template. To streamline this task, we propose a JSON-based file format to specify a base sewing pattern as a structure alongside a set of rules describing allowed changes on the panels' edges and the set of constraints on edges to keep (Section 3.1). The proposed format allows efficient and uniform descriptions of design spaces of garments that share the same sewing pattern topology.

The rest of the pipeline is fully automatic. A dataset generator automatically draws a requested number of random sewing patterns from provided template specification, and then each of the generated sewing patterns is simulated on top of a human body model to create draped 3D garment shape (Section 3.3). To bridge the gap between clean simulated meshes and typical in-the-wild data, each 3D garment can be additionally post-processed. This last step imitates the artifacts of the typical 3D scanning process by removing parts of the surface that are not visible from the outside view (e.g., occluded by folds), as described in Section 3.4.

The datasets obtained with our pipeline have a well-controlled design distribution, with the datapoints being well-aligned both in 3D space and in sewing pattern spaces. Extending and creating new datasets comes down to designing a new template which can be done as needed.

Using the designed pipeline, we created a large-scale dataset of garment 3D models with thousands of garment design samples, coming from 12 template garments that target model training and 7 templates to evaluate the generalization across sewing pattern topologies (Section 4). We hope that the community will help expand the dataset of garment designs even further by creating new garment templates. The dataset, along with the base templates, is available on Zenodo [1]. Code for the generator is available on GitHub[1].

## 2   Related Work

This section gives a brief overview of the existing datasets related to 3D garments.

Few publicly available datasets provide garment patterns alongside 3D garment models. Berkeley Garment library [11, 12] provide sewing patterns geometry, but consists of only 21 garment examples of different garment type and thus not suitable for training Deep Learning models that aim to generalize across garment designs as in our work. Wang et al. [13] released part of their training dataset publicly, and it contains both 3D models and garments but only offers one garment type (sweaters) with a simplified pattern topology that consists of a single panel. Vidaurre et al. [14] describe a dataset of 19 garment examples with corresponding sewing patterns, all of them sharing the same pattern topology. Our data generation pipeline allows obtaining large-scale datasets suitable for training Deep Learning models and covering various garment types and designs, as demonstrated by our dataset.

Other notable datasets of 3D garments that do not provide sewing pattern information include datasets constructed from real scans, such as BUFF [15], Multi-Garment Net [6], CAPE [7], Deep Fashion3D [16], and SIZER [17], and synthetic datasets, such as 3DPeople [18], BCNET [19], and CLOTH3D [20]. In terms of design variation, Deep Fashion3D [16] is the largest among the ones based on real data featuring the current state-of-the-art 563 diverse scanned clothing items. However, it is still a rather sparse sample of the garment distribution and hence does not eliminate the need for

---

[1]https://github.com/maria-korosteleva/Garment-Pattern-Generator

obtaining design variations synthetically. As for synthetic datasets, 3DPeople [18] uses unique outfits for each of 80 human characters, BCNET [19] rely on variations of six base garment types (short and long templates of upper garment, pants, and skirts), while CLOTH3D [20] relies on shape variations and combinations of four template garments (t-shirt, top, trousers, and skirt). Our generation pipeline gives means for even larger design diversity by introducing template definition language. Our dataset relies on 19 base types, but most importantly, it could be extended to include new base garments within the same framework without additional effort in development.

To the best of our knowledge, none of the existing synthetic garment datasets address the gap between clean artificial meshes and the noisy geometry of in-the-wild scan data. We provide an imitation of some of the artifacts met in the raw data to assist with bridging this gap in the downstream tasks.

As for the data generation process, our work shares the approaches of [13, 14, 19] that sample datapoints from a parametric sewing pattern and then drape them on the target body. However, we take the idea further by creating a language for defining parametric templates to simplify the generation of new synthetic datasets of garment designs and expand the design spaces. CLOTH3D [20] is one of the closest works to ours as it provides not only a synthetic dataset but also a data generation method to vary garment designs automatically. However, these design variations are introduced in 3D by cutting and re-arranging the simulated garment models, which allows fast operations but does not guarantee physically correct garment drapes. In contrast, our work introduces variations in sewing pattern space before applying physics simulation, hence providing realistic shapes.

## 3  Dataset Generation System

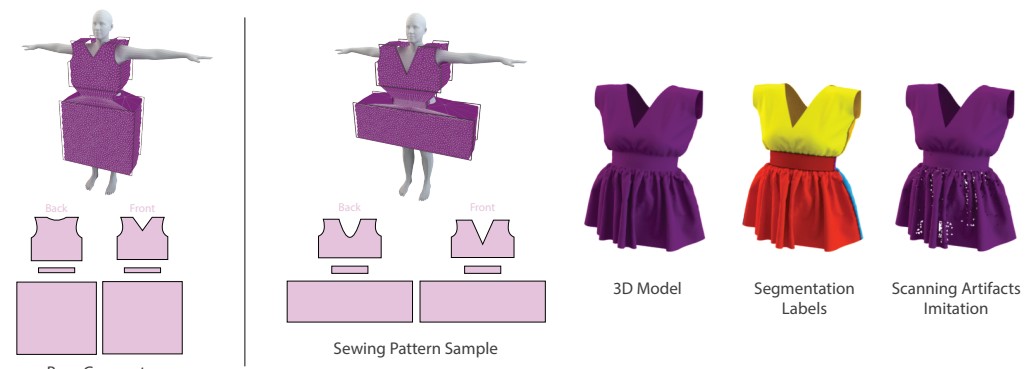

Figure 2: Example data instance. Our data generation pipeline samples the design space of the base sewing pattern and drapes the sample over a body to get the corresponding 3D model. We also generate a segmentation map of this model (different colors correspond to distinct panels) and an optional version of the 3D model that imitates 3D scanning artifacts.

This section presents our dataset generation pipeline. In order to make the dataset generation easily extensible, our system follows the classical software design principle of Separation of Concerns (SoC), in our case separating the definition of parametric sewing pattern templates from garment-type agnostic dataset construction. This approach means that creating a dataset for a new garment type only requires the specification of a new template without a need to change the system itself.

For a convenient and flexible definition of sewing pattern parametric templates, we present a domain-specific file format (Section 3.1). The manual template development process is further supported by a GUI (Figure 6), where one can visualize the defined model and try out the behavior of a template through all stages of the dataset generation.

The automatic dataset construction involves template sampling (Section 3.2), draping the sewing pattern samples on provided body model (Section 3.3), producing 3D mesh and its segmentation into pattern panels, and scan imitation process that removes parts of the hidden surface when viewed from outside (Section 3.4). The computational performance of our automatic stage depends on many factors, from hardware setup to simulation configuration and garment complexity, but we report the values for our dataset in Section 4 to provide a reference.

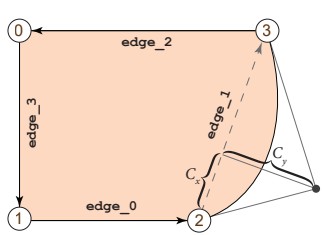
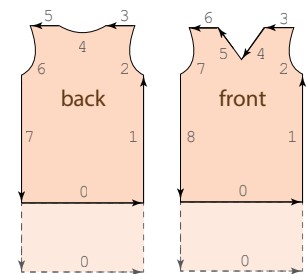
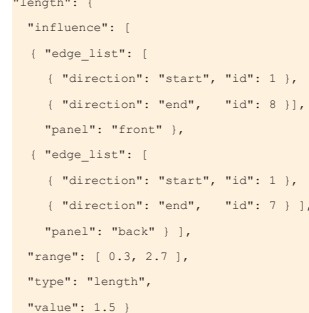

```
"length": {
  "influence": [
    { "edge_list": [
      { "direction": "start", "id": 1 },
      { "direction": "end",   "id": 8 }],
      "panel": "front" },
    { "edge_list": [
      { "direction": "start", "id": 1 },
      { "direction": "end",   "id": 7 } ],
      "panel": "back" } ],
  "range": [ 0.3, 2.7 ],
  "type": "length",
  "value": 1.5 }
```

Figure 3: Panel representation. The edges form an oriented loop and are labeled sequentially. $(C_x, C_y)$ define the location of quadratic Bezier control point for the curvy edge *edge_1*, expressed relative to the edge.

Figure 4: Parameter rule to control pattern length. It acts on the edges length (hence the type *"length"*). The change is uniformly applied to both panels on their side edges (*"influence_list"*). The direction of application instructs to extend towards the lower part of the garment. The current value of 1.5 shows that the edges are to be extended to 1.5 times their base length (New shape is indicated as a lighter area on the pattern).

Note that in this work, we are specifically interested in exploring designs of garment shapes, and thus the pipeline is not currently optimized for automatic sampling of material properties, body shape, or pose variations. These parameters can be set freely before the generation process but remain fixed thereafter.

The code of the data generator will be available for free for research and other non-commercial purposes.

## 3.1 Pattern Template Specification

A sewing pattern template specification is a JSON-based file format with a pre-defined domain-specific structure. A specification consists of the base sewing pattern, parametrization, and, optionally, constraints to be held while sampling parameter space. We chose JSON as a basis for our specification because it is simple, human-readable and editable, and structured while still convenient for automatic processing. Example template specifications are provided with the dataset that accompanies this paper (see Section 4 for dataset description).

### 3.1.1 Base pattern definition

The first part of the template specification describes a base sewing pattern that would then be parametrized and varied to produce new designs. The structure for the pattern description has been designed to reflect the general definition of a garment sewing pattern as closely as possible to allow a variety of garments to be described with it, even those that we may not foresee. Nonetheless, we added a simplifying restriction that the edge curvatures need to be either linear or quadratic at this stage of development. We also require edges to be oriented and defined sequentially to form a loop (see an example in Figure 3, which is a rather natural way of describing closed polygons. This requirement gives useful assumptions to rely on in parameter rules specification (see Section 3.1.2).

Hence, the structure for specifying a base sewing pattern consists of the following components:

- **Unordered list of panels.** Every panel is, in its turn, a multi-component structure described with the following elements:
  - A list of 2D coordinates of panel vertices, specified in local space of the panel. The order can be defined arbitrarily.
  - An ordered list of oriented edges that form a loop. Each edge is described by the IDs of two vertices it connects and 2D coordinates of the quadratic Bezier curve control point (named *curvature coordinates*) if the edge is not a straight line. The coordinates are specified relative to the edge (Figure 3).

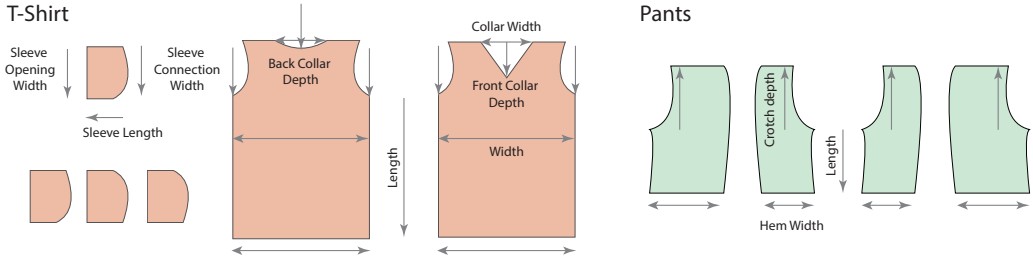

Figure 5: Parametrization for two of our base sewing patterns. The pants template has three parameters defined, while a more complicated T-shirt template defines nine parameters (Note that the sleeve parameters are applied equally to all sleeve panels).

- – Global translation and rotation (Euler angles) that define panel location in 3D around a particular body model, which would then be used for draping.
- **Unordered list of stitches.** Every stitch is defined as a pair of edges to be joined. Edges are references by the panel name and edge ID in the definition of that panel.

### 3.1.2 Parameter rules

Our goal with pattern parametrization was to create a rule definition language that is expressive enough to represent a variety of design variations, allowing symmetric and asymmetric changes, garment parts varying simultaneously (e.g. left and right sleeves) or independently from each other, multiple variations to the same base pattern or even to the same area of the base garment. On the other hand, we want the rule language to be structured simply and to avoid heuristic assumptions on pattern topologies, such as expecting certain components to be present or absent.

With these considerations in mind, we devised an approach to defining rules at the panel edge level. An advantage of using such a basic building block is that the scope of its possible variations is concisely defined comparing to higher-level concepts like panels. Changes of length and curvature of individual edges can be composed into semantically meaningful design variations (e.g., extending sleeve length is equivalent to extending the edges located alongside the arms). Defining rules for different parts of garments is simplified to picking the set of edges from the panels of corresponding semantic parts. Additional flexibility is achieved by controlling the way an edge property varies (e.g., toward the start or the end of the edge, along the edge itself or some other 2D vector).

Varying the panel vertex positions directly was another option to consider, but it has some limitations. Vertices themselves do not hold information of edge curvatures, which prevents the direct control of edge curvatures. Direct access to edges directions is also highly useful as many variations are naturally defined along edges in the garment sewing patterns.

The final proposed structure of parameter rule is composed of the following attributes.

- *List of edges* from any of panels that a particular parameter influences. These edges are modified uniformly upon parameter sampling to enable symmetric changes when needed.
- *Target property* – curvature coordinates or edge length. Changing curvature coordinates by parameter value is straightforward as it is a local property of an edge. The edge length parameters, by default, create changes along the edge's main direction by modifying the edge's 2D vertex positions with respect to the local space of the panel. To allow more flexibility, changes can be constrained to occur along an additionally specified 2D vector, e.g., only vertically or to be distributed across a sequence of contiguous edges.
- *Parameter type* – multiplicative or additive. The target property is changed by multiplication by or addition with the parameter value.
- *Allowed range of values.* Upon sampling, a new parameter value is drawn from this range.
- *Current parameter value.* For templates, it defaults to 1 in multiplicative and 0 in additive parameters.

An example definition of a parameter rule is given in Figure 4. Figure 5 shows examples of complete sets of parameters for two base garments.

The order of applying multiple parametric rules may influence the result, and thus the specification includes explicit definition of parameter order to ensure sampling consistency and predictable results.

### 3.1.3 Constraints

In addition to the parameter rules, the template allows the definition of constraints. *Constraints* are the rules that help restore desired consistency of the sewing pattern stitches by forcing the edges on both sides of the stitch to have the same shape. Thus, regardless of how the template parameters are defined, the constrained edges will change together. The need for defining constraints depends on base sewing pattern shape, parameter deviation, and desired outcome. When preparing our dataset (Section 4), we found that for many examples, it was possible to define parameters in the way that stitch consistency is never broken even without constraints, except for one of the skirts templates (*Skirt 4 panels*).

A constraint is specified as a list of affected edges and a per-edge modification value (defaults to 1 in templates). Upon application, constraint evaluates the average length of the specified edges and clamps or extends every edge to this average length. Per-edge modification values store the amount of change applied as a multiplier of the edge length, allowing to restore the original state if needed.

## 3.2 Sampling sewing patterns

With a pattern template defined, the next step is to sample individual patterns from the template. This process is relatively straightforward. The predefined parameters are applied to the base pattern one by one in the specified order: a random value sampled from the specified range is applied to the list of edges according to the parameter and property types. Afterward, if any are defined, a list of constraints is applied to a sample. With the flexibility of our parameter rules, some of the samples may contain topological errors, specifically self-intersecting panels. Thus, every panel of the newly generated pattern is examined for self-intersections, and if detected, the pattern is discarded, and a new sample is drawn instead. The final sample is saved in a JSON file with the same structure as the original template but with updated sewing pattern definition and parameter values.

In principle, the choice of the panel order within a sewing pattern, and the choice of the first edge in an edge loop of panels, can be arbitrary for defining a sewing pattern shape. However, ensuring predictability and consistency of these properties across the data samples can be beneficial on the application side and would be hard to obtain if not introduced during generation. Hence, we first pre-process the base pattern defined in the template: the panels are sorted ac-

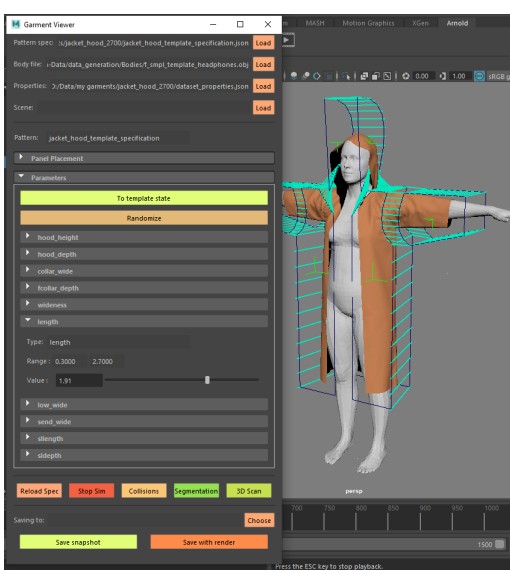

Figure 6: Our GarmentViewer GUI for testing sewing pattern templates (developed within Autodesk Maya framework). It supports loading a pattern as a mesh, changing parameters, testing simulation, segmentation, and scan imitation on the loaded garment, and saving the current garment state.

cording to their 3D translations, in the order of $X$, $Y$, and $Z$ coordinates; all edge loops are ensured to follow counterclockwise direction; for every panel, we pick an edge that originates from the lowest-leftmost vertex as the first on in the edge list. This ordering is then propagated to all the samples from this template. Picking the order by rules rather than using an arbitrary order made by a template creator gives better consistency across different templates and, as a result, in datasets constructed from multiple templates.

### 3.3 Obtaining garment 3D model

After sewing pattern samples are obtained, each is draped on top of a human body model using physics simulation. Generally speaking, any 3D object can be used as a body model, but it is recommended to use the same body model that the template panels were placed around (see Section 3.1.1 for more on placement) to increase the simulation stability. Material properties for both fabric and body model and the desired mesh resolution can be set freely for every sample batch. Final models are saved as OBJ files. In our work, we choose Qualoth as the base physics simulator thanks to its robust production quality results and the ability to work with garment sewing patterns as curve collections, saving us development time.

The final 3D models are accompanied by per-vertex segmentation labels indicating the panel each vertex belongs to or whether it belongs to the stitch (see example visualization in Figure 2). These values are propagated from an initial sewing pattern through the simulation process. Segmentation labels are stored as a list in the TXT file.

Every simulated garment mesh is post-processed to ensure the high quality of the output results. With high flexibility in defining design spaces, some sewing pattern samples may produce 3D shapes that penetrate the body model or have self-intersections. If these problems are not resolved during physics simulation, the problematic sample is marked as failure, so that the sample can be easily excluded from consideration in downstream tasks.

It is also important to note that the meshes produced in this step do not share the same mesh topology but are well-aligned in 3D in terms of scaling and global placement.

### 3.4 Imitating 3D scanning artifacts

The garment meshes produced by the simulation step are clean and describe the garment shape completely, which is very much unlike the data captured in-the-wild by 3D scanning or other means. To assist the downstream tasks with generalization to such data, we add a post-processing step to imitate mesh imperfections of the wild geometry. More specifically, we observe that most of the means of obtaining 3D models would miss the regions not visible from outside cameras, e.g., occluded by the garment's own folds or body parts. We imitate such artifacts on the 3D models produced by the simulation step, as in the example shown in Figure 2.

To find occluded areas on an arbitrary garment, a draped garment 3D model and a body model are placed in a tight box with an empty floor and ceiling, representing a 3D scanner. Every face of a garment mesh is tested on visibility from the walls of this box. The visibility test is performed by simply shooting a bunch of rays in random directions from the center of a face and checking if a given fraction of rays hit the walls. The invisible faces are removed, as well as any vertices that have all of their adjacent faces removed. The fraction of visible rays can be specified during data generation, but we found that the value around 10% produces reasonable results. The segmentation labels of the original mesh are also propagated to the corrupted meshes.

Other types of data acquisition artifacts, such as irregular mesh topologies, vertex position noise, or missing parts due to a limited number of cameras, are not included in the pipeline. We believe that these effects depend on the task at hand and would be fairly easy to imitate as needed.

## 4 Our Dataset

With the generation pipeline presented in Section 3, we created a large-scale dataset of 3D garments with sewing patterns, which is made publicly available under CC BY 4.0 on Zenodo [1]. The complete documentation is provided as Datasheet [21] that accompanies the dataset, and this section presents an overview.

The dataset contains a total of 23500 garment design samples, among which 22547 passed the simulation quality checks as described in Section 3.3 and thus are safe to use in the downstream applications. To create this dataset, we designed 19 sewing pattern templates, examples of which are given in Figure 7. The data samples can also be found in Figure 1 and throughout the paper.

The 19 garment templates are divided into training and test groups. The training group of 12 templates aims to cover design spaces of typical simple garments, including skirts, dresses, tops, pants, jackets,

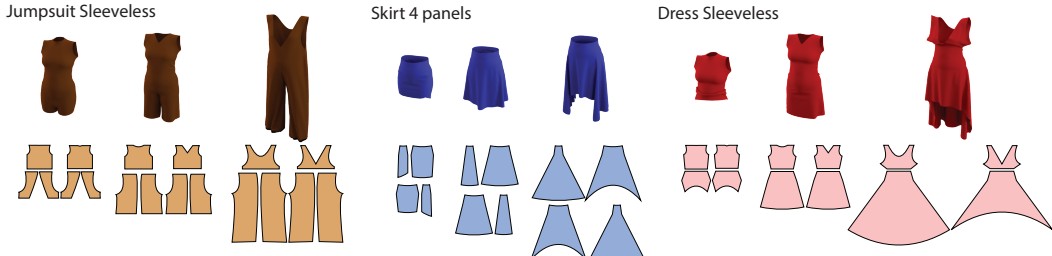

Figure 7: Three of 19 templates that were used for generating our dataset. Each template is represented as the base sewing pattern and its 3D model (in the middle of each group), and two extreme variations of the base – with most parameters set to minimum values (left) and most parameters set to maximum values (right). Visualizations of all 19 templates are provided in Figures A1 and A2 in Appendix.

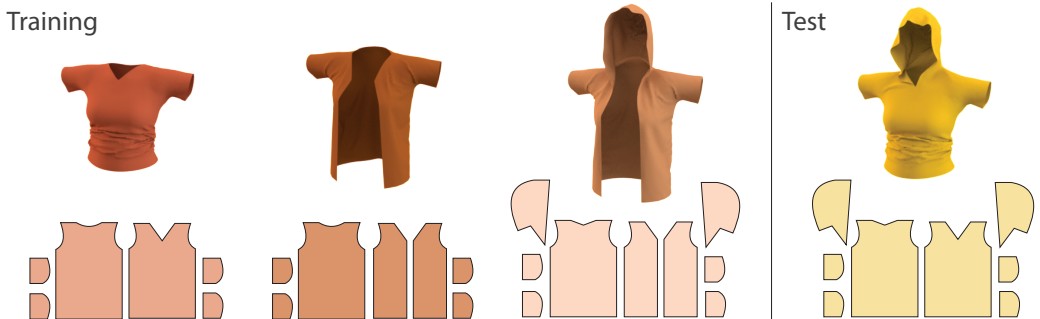

Figure 8: Example of a base garment from the test group composed from the same panels as base garments of the training group but arranged into a novel sewing pattern topology. This example evaluates the capability to reconfigure a t-shirt into having a hood similarly to the open jacket examples.

hoodies, and jumpsuits, and reflect some topological variations among them. We draw 1000 to 2700 samples from each training template depending on the complexity of the template parametrization. The number of parameters in the templates varies from just 3 (e.g., in pants) to 11 (in a sleeveless dress). Seven templates from the test group target the evaluation of generalization across sewing pattern topologies. Base patterns are designed strictly from the same parts as the garments from the training group, with the same parametrization, but these parts are re-arranged to give new topologies that are not directly included in the training group, as illustrated in the example in Figure 8. We draw 150 samples from each template of the test group. Note that the primary goal of the suggested training/test division is to prevent some garment types from being used during training and only appear at evaluation time. In practice, we recommend including some portion of samples from 12 training templates into the test set for an unbiased evaluation.

The template design and simulation process require a choice of body model and material properties. We use an average female body model provided by SMPL [22] in T-pose and a fixed set of material properties for all of the templates and data samples we created. All of the garment samples have both clean meshes and their versions with scanning artifacts.

## 4.1 Computation performance

The time required to design the template, the manual part of the work, ranges from few hours to few days depending on the complexity of the base garment and its parametrization, whether parts of other templates could be reused, and the familiarity with it the process.

Compute time for the automatic stage of the dataset creation varied depending on the hardware, simulation settings, and garment mesh vertex count. The sewing pattern sampling stage is the least expensive, with about 1300 designs generated per minute. For each garment, it took about 3 minutes for simulation, 45 seconds for the scan imitation, and 1 minute for creating two rendered images

for visualization. The total time to produce the dataset would be 47 days for physics simulation, 12 days for scan imitation, and 15 days for rendering if all garment samples were sequentially processed. Parallel processing is possible for the dataset generation and reduces the time significantly. Reported values were obtained on machines with either Intel Core i5-8500 CPU or AMD Ryzen 9 3950X CPU and a single NVIDIA Titan XP GPU used by the renderer.

## 4.2 Potential usages of the dataset

We provide a highly consistent dataset that specifically targets garment design variations. The dataset presents several challenges for the Deep Learning research, including the following:

- Modeling complex structures like sewing patterns, which are essentially sets of variable length consisting of variable structured parts (panels) with additional cross-references among panels' components (as stitches are defined on edges);
- Understanding the construction principles of these sewing patterns and generalizing across topologies;
- Working with structured *deformable* objects.

These challenges would arise in both the cases when sewing patterns are used as inputs (e.g., generating draped shapes) or outputs (e.g., estimating garment structure from meshes).

## 5  Discussion and Future Work

This paper introduces a data generation pipeline with a flexible method to define parametric sewing pattern templates, followed by an automated pipeline of template sampling, batch simulation, and scanning artifacts imitation modules. The paper is accompanied by a large-scale dataset [1] of more than 20000 garment designs with sewing patterns and draped 3D garment meshes. The dataset was synthetically generated from 19 originally designed garment templates.

**Limitations**   There is still much to be done to create datasets representing various garment designs and diverse ways people wear the garments. The proposed data generation pipeline is largely simplified by fixing material properties, body pose and shape, and omitting fine features of sewing pattern design, such as darts, pleats, or complex edge curves. We plan to enrich the current pipeline to cover these variations and provide a broader range of base parametric templates. On top of that, it would be interesting to reflect complex garment arrangements, such as layering of fabric (e.g., ballroom skirts), use of accessories, overlaying of multiple garments, and utilization of complex materials like thick and bumpy winter coats.

**Ethical concerns**   We do not see significant risks of security threats or human rights violations in our work or its potential applications. However, we do foresee that our work contributes to the field of garment analysis and virtual garments development fields overall. These efforts might eventually remodel the clothing production, retail, and virtual garments development, leading to changes in the workforce structure. Hence, there is a general concern that some jobs might be automated fully or significantly to reduce the demand for human workers, and the industries would need to act proactively to avoid the social impact of such changes. Another issue that concerns us is the potential for intellectual rights violations. Models build to evaluate garment structure might be misused to reverse-engineer original works of garment design, simplifying their unauthorized reproduction. The current regulations in this area might need to be modified to include such technological advances.

## Acknowledgments and Disclosure of Funding

We would like to thank the anonymous reviewers for their helpful suggestions. This work was supported by National Research Foundation, Korea (NRF-2020R1A2C2011541), and Korea Creative Content Agency, Korea (R2020040211). Autodesk and FXGear provided research licenses to their software packages, Autodesk Maya and Qualoth, which helped us greatly speed up the development of our pipeline. The members of Geomteam of Motion Computing Lab, KAIST, deserve our deepest appreciation for the invaluable discussions and moral support throughout this project.

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
