# OpenReview forum: "Generating Datasets of 3D Garments with Sewing Patterns"
_NeurIPS.cc/2021/Track/Datasets_and_Benchmarks/Round1 — NeurIPS 2021 Datasets and Benchmarks Track (Round 1)_

### Official Review · Reviewer_tBLx · 2021-07-03
**A large dataset for 3D garments with sewing patterns**

**Rating:** 7
**Confidence:** 3
**Correctness:** Correct.
**Clarity:** Yes.

**Strengths:**

1. Detailed introduction for how to generate the dataset.
2. The generated dataset contains a lot of 3D garments, which provide great potential for future research.


**Weaknesses:**

1. Paper contains some gramma and spelling errors, eg.,
Line 23   work -> works
Line 65   describes -> describe
Line 310 modeling -> modelling



**Additional Feedback:**

See comment above.

**Documentation:**

It provides documentation for data collection, organization, availability and other information.

**Ethics:**

No.

**Relation To Prior Work:**

Clearly discussed.

**Summary And Contributions:**

The authors proposed a pipeline to generate the 3D garments dataset for many applications like modelling complex structures and decouple the clothes to sewing patterns. The generated dataset contains 23500 garment design samples with comprehensive information for each samples.

---

> ### Author Response · Authors · 2021-07-12
> **Thank you!**
>
> Thank you very much, we highly appreciate your feedback! We fixed the typos you mentioned and will proofread the paper again before publication.

---

### Official Review · Reviewer_GXpx · 2021-07-05
**An interesting 3D Garments dataset**

**Rating:** 6
**Confidence:** 3

**Strengths:**

- It is an interesting and novel dataset we have never seen but will find useful. Recently there are a bunch of papers about 3D humans but few of them focus on the clothes. Starting from a large-scale synthetic dataset sounds a cool idea to me.
- The dataset even imitates artifacts of 3D scanning. I would expect it to be helpful when the model is generalized to real-world 3D scans.
-  I appreciate the data organization. I can easily look at the rendering of each clothes.

**Weaknesses:**

- The paper does not show any practical usage of the dataset. This is my main concern of the paper. I would expect training a human pose model including clothes on the dataset and show some generalization to real-world data. From a synthetic dataset it is hard to evaluate whether it will generalize to real images by looking at rendering results.
- In practice, the dataset may be biased towards women since it is using an average female SMPL body model. It raises more concerns about the dataset when no experiments are shown to demonstrate the usage of the dataset.

**Additional Feedback:**

Typos

L306 "Tytan" -> Titan

**Clarity:**

The paper is well written.

Update: The paper is further updated to clarify points made by other reviewers.

**Correctness:**

The dataset is constructed in a sound way. A lot of details are provided in the supp.

**Documentation:**

- The supp provides a QA for all dataset details. It is expected to be released before Sep 2021 and will be maintained and developed in the future.
- There is also a dropbox url to view the the dataset. I've checked the rendering of examples and it looks good to me.

**Ethics:**

This is a synthetic dataset so I cannot foresee any ethics concerns.

**Relation To Prior Work:**

The paper is the first large-scale dataset of 3D garment models. So we cannot find direct comparison with previous contributions. Related works include several small-scale datasets such as  Berkeley Garment library, which is discussed in section 2. Obviously the paper is significantly different from them.

The paper may want to cite more approaches about human cloth capture such as MonoClothCap [1].

[1] Donglai Xiang, Fabian Prada, Chenglei Wu, Jessica Hodgins. MonoClothCap: Towards Temporally Coherent Clothing Capture from Monocular RGB Video. 3DV 2020.

**Summary And Contributions:**

The paper proposes the first 3D garment dataset. Given there is no existing datasets for 3D garments and rising interests of researchers, the paper creates a semi-automatic pipeline to generate large synthetic datasets of 3D garment designs and their sewing patterns. The pipeline also generates corrupted version of meshes to imitate the artifacts of 3D scanning.

---

> ### Author Response · Authors · 2021-07-12
> **Thank you!**
>
> Thank you very much for your feedback! We addressed the question on the experimental results in the overall answer above.
>
> As we acknowledge in the paper, our dataset relies on some assumptions (including pose and type of body) that would likely prevent it from generalizing to real-world data. Our goal was specifically to bring a larger variety of garment designs to the field, but we hope to address generalization in other directions in future work.
>
> Thank you for your suggestion on additional citations, we've added it to the paper.

---

> > ### Comment · Reviewer_GXpx · 2021-07-21
> > **updated review**
> >
> > Thanks for your clarifications! Overall I believe it will be of value to NeurIPS. It seems that other reviewers also agree on this. I'm going to keep my original rating 6.

---

### Official Review · Reviewer_7Mpt · 2021-07-05
**An interesting paper on generating 3D garment dataset**

**Rating:** 7
**Confidence:** 5
**Clarity:** The paper is well-written and easy to…

**Strengths:**

1. The presented dataset constitutes a solid foundation for further garment modeling research, especially in a deep learning framework.

2. The pipeline is natural and understandable. The generation process is based on 2D patterns, allowing for additional designs to be implemented efficiently.

3. The authors also provide segmentation labels for the mesh vertices, making this dataset appropriate to construct learning algorithms for garments.

**Weaknesses:**

1. Although the authors include segmentation labels for their data, they do not provide any experimental evaluation on their dataset to benchmark and illustrate their approach's usefulness.

2. Train/test split predefined by the authors implicates designating several distinct patterns into a test set. There is a concern that such a strategy would introduce an implicit bias. For instance, there are no pants-like categories in the test split.

3. Usage of a proprietary software (Autodesk Maya) may be limiting for future research.

4. The scanning simulation process is basic and does not reflect any real scanning noise or corruption. Although the authors acknowledge this in their paper, the proposed process and simulation results are not of great use. Also, from its description, it appears that scanning simulation is discarding mesh triangles but leaves mesh vertices as is. Since segmentation labeling is per-vertex, this process has virtually no effect on the training data.

**Additional Feedback:**

1. The benchmarking study is always a welcome addition to the dataset, at least in future work.

2. Careful train/test split consideration is required; possibly a random split is more meaningful.

**Correctness:**

The data generation process is transparent and meaningful. The presented resulting samples and code implementation suggest that the proposed approach is reasonable.

**Documentation:**

The authors provided a link to the generated samples with an implementation of the generation pipeline. Documentation in supplementary materials fully describes the structure of the dataset, diversity of the models, and future maintenance plan.

**Ethics:**

No direct ethical concerns are spotted.

**Relation To Prior Work:**

The authors cite and describe the existing datasets of garments and clearly indicate the difference. The proposed dataset has more instances, considerable diversity, and an automated design pipeline, distinguishing this paper from the previous work.

**Summary And Contributions:**

This paper proposes a pipeline to generate 3D garments from 2D patterns. This approach is primarily automatic, requiring minor manual interventions. It is defined naturally similarly to the way real-world patterns are stitched during garment production. The dataset can help improve several tasks in deep learning framework, including garment simulation and understanding.

The main contributions include the dataset pipeline with the implementation of all stages and the dataset generated by the authors consisting of more than 20,000 samples.

---

> ### Author Response · Authors · 2021-07-12
> **Thank you!**
>
> Thank you very much for your detailed review! We addressed the lack of experimental evaluation in the overall answer above. As for the test\train split by type, this split specifically targets the question of generalization across pattern topologies. It might not be suitable for all applications, and for now, we left the decision of making a split to the dataset users. We added clarification on this matter to Section 4, and improved consistency of using "test groups" versus "test set" throughout the paper.
>
> The last note is on the scan simulation. Actually, all the "hanging" vertices that have all their adjacent faces removed are also removed. We clarified this in the updated paper.

---

### Author Response · Authors · 2021-07-12
**Thank you for reviews and note on the experimental evaluation**

We would like to thank all our reviews for their valuable evaluation and feedback. Your time and effort to do this important job are sincerely appreciated!

We updated the paper with some clarifications (Section 3.4, Section 4, reflected in red color) and fixed typos according to the suggestions in reviews.

We would like to address the concern on the lack of usage example \ benchmark results. As was suggested in the paper, our dataset is suitable for multiple different tasks. A thorough benchmarking of those would require substantial additional effort. While still working towards some of these tasks, we wanted to share the results we have so far with the community to promote further research and exploration of the 3D garments domain. But we fully agree that benchmarking study is an important step, and we hope to address it in future work.

---

### Decision · Program_Chairs · 2021-07-26

**Decision:**

Accept

**Comment:**

The paper provides a 3D garment generator that can be very useful for the clothing, fashion, and 3D human analysis community. This looks to have a unique value compared to the very few existing alternatives. Overall reviewer opinions are toward acceptance of the paper.